# Humidification during Invasive and Non-Invasive Ventilation: A Starting Tool Kit for Correct Setting

**DOI:** 10.3390/medsci12020026

**Published:** 2024-05-15

**Authors:** Riccardo Re, Sergio Lassola, Silvia De Rosa, Giacomo Bellani

**Affiliations:** 1Anesthesia and Intensive Care 1, Santa Chiara Hospital, APSS, Largo Medaglie d’Oro 9, 38112 Trento, Italy; riccardo.re@apss.tn.it; 2Centre for Medical Sciences—CISMed, University of Trento, Via S. Maria Maddalena 1, 38122 Trento, Italy; silvia.derosa@apss.tn.it (S.D.R.); giacomo.bellani@apss.tn.it (G.B.)

**Keywords:** active humidification, heated humidifier, pass-over humidifier, passive humidification, HME, ICU

## Abstract

The humidification process of medical gases plays a crucial role in both invasive and non-invasive ventilation, aiming to mitigate the complications arising from bronchial dryness. While passive humidification systems (HME) and active humidification systems are prevalent in routine clinical practice, there is a pressing need for further evaluation of their significance. Additionally, there is often an incomplete understanding of the operational mechanisms of these devices. The current review explores the historical evolution of gas conditioning in clinical practice, from early prototypes to contemporary active and passive humidification systems. It also discusses the physiological principles underlying humidity regulation and provides practical guidance for optimizing humidification parameters in both invasive and non-invasive ventilation modalities. The aim of this review is to elucidate the intricate interplay between temperature, humidity, and patient comfort, emphasizing the importance of individualized approaches to gas conditioning.

## 1. Introduction

The integration of heating and humidification in medical gases has been a standard process since the early adoption of mechanical ventilation in clinical practice. In one of the earliest prototypes of positive pressure ventilators, such as the Morch’s respirator (1954), a humidifier was already incorporated [1,2]. Indeed, the necessity to heat and humidify gases in ventilated patients became evident early due to the observed injuries caused by compressed medical gases during invasive mechanical ventilation [3].

The requirement for gas warming and humidification arises from the use of invasive airway management devices such as endotracheal tubes, tracheostomies, and supraglottic systems. These devices bypass the initial airway section, disrupting the natural physiological process of gas heating and moisturizing [4]. This becomes particularly pertinent as medical gases are stored and dispensed at low temperatures and devoid of humidity [5]. Without the proper humidification and heating of medical gases, complications associated with bronchial dryness can rapidly manifest such as mucosal injuries, a decrease in muco-ciliary clearance, leading to the retention of secretions, infections, and the obstruction of ventilatory devices [6,7]. Additionally, the production of surfactant can be adversely affected. This situation contributes to a reduction in functional residual capacity (FRC) and lung compliance, characterized by a decrease in alveolar diameter and lung compliance [8]. Extended exposure to cold gases during ventilation can also induce hypothermia in the patient [9]. Conversely, it is essential to note that an incorrect humidifier configuration, with excessively high temperatures and humidity levels, can result in functional and anatomical alterations such as mucosa thermal injury, worsened mucus rheology, and dysfunction of muco-ciliary clearance. Excessive humidity can also compromise surfactant functionality, leading to micro-atelectasis with a reduction in FRC and lung compliance [1].

The importance of humidifying medical gases during ventilation is well established in Intensive Care Units (ICUs) [3]. In the past, aerosol-based devices were commonly used. Instead, the current trend favors both active and passive humidification systems in ICU. Active systems include adjustable electrical humidifiers, while passive systems involve filters capable of capturing heat and moisture during the expiratory phase and redistributing it in the inspiratory phase [10].

For several years, it has been known that both the upper and lower airways condition the inspired air to reach a temperature of 37 °C [11]. Despite this knowledge, determining the right humidifier setting for patients undergoing invasive ventilation has been a debated topic. Initial recommendations ranged between 25 and 30 °C, but subsequent scientific research led to a shift in recommended temperature targets to 36–37 °C, endorsed by current ventilation guidelines [12]. Despite this knowledge in invasive ventilation, the humidification process is often underestimated. This underestimation becomes even more critical in non-invasive ventilation. Although humidification is sometimes deemed unnecessary in non-invasive ventilation, it is still recommended [1,12]. The target is an inspiratory gas with absolute humidity of 10 mg/L. However, achieving this target using medical gases in non-invasive ventilation is not always straightforward [13,14,15].

The intricate nature of the humidification process is often underestimated by clinical operators [16]. The aim of the present review is to offer a technical and practical guidance to optimize medical gases heating and humidification management during invasive and non-invasive ventilation.

## 2. Physical Aspects

### 2.1. The Humidity

Absolute humidity measures the actual amount of water vapor present within a specific volume of gas. It is usually expressed in kilograms per cubic meter (Kg/m^3^) or in milligrams of water per milliliter of air (mg/mL) in the ICU context (Figure 1, Panel A) [17]. On the other hand, relative humidity is the ratio between the absolute humidity (the actual amount of water vapor in each volume of gas) and the maximum amount of water vapor that the same volume can hold. Relative humidity can also be expressed as the percentage of water “availability” in each system. Moreover, absolute humidity is influenced by temperature and pressure. Indeed a gradual temperature decrease increases relative humidity until the ‘saturation point’ (100%), where air reaches its maximum water-carrying capacity. A further temperature decline leads to the ‘dew point’, causing water condensation and a drop in absolute humidity. Below the ‘dew point’, temperature no longer affects absolute humidity. Under isothermal conditions, an increase in absolute humidity corresponds proportionally to higher relative humidity [17,18].

In Figure 1, Panel A illustrates the relationship between humidity and temperature, highlighting the saturation point. An increase in temperature enhances the air’s capacity to carry water vapor (absolute humidity), thereby lowering the relative humidity. In particular, in this example, at 20 °C, the room air is saturated (RH 100%) with 18 mg/L of water vapor. However, when the temperature increases to 30 °C, maintaining the same 18 mg/L of water vapor, the relative humidity drops to 50%. Panel B, consider a hypothetical isolated system with 10 mg/L absolute humidity. At 30 °C, the gas has 50% relative humidity. A drop to 20 °C increases the relative humidity to 75%, holding absolute humidity at 10 mg/L. Further reduction to 10 °C leads to 100% relative humidity, marking saturation point with constant 10 mg/L absolute humidity. Lowering the temperature to 4 °C brings the dew point, causing condensation. Relative humidity stays at 100%, but absolute humidity drops to 5 mg/L. The continued temperature decrease to 0° C results in complete water vapor condensation, yielding 0 mg/L absolute humidity and 0% relative humidity [17].

### 2.2. Humidification Target Values in Clinical Practice

In the context of invasive ventilation, the international literature commonly recommends maintaining a relative humidity of 100%. This corresponds to ensuring that the gas delivered to the patient, typically at the Y-connection of the ventilatory circuit, contains 44 mg/L of water vapor at a temperature of 37 °C. These parameters emulate physiological conditions, ensuring proper moisture levels and preventing complications associated with dry airways [19,20,21,22,23]. While the importance of humidification is well acknowledged in invasive ventilation, its application in non-invasive ventilation is often overlooked. In the latter, the initial airways are not bypassed, preserving the natural process of heating and moisturizing gases. However, the nature of the gas (dry medical gas vs. room air blending) and the treatment duration must be carefully considered. The absence or suboptimal provision of humidification over the long term can lead to complications such as mucous membrane injury, odynophagia, conjunctivitis, and dehydration [1,13]. In particular, devices utilizing oxygen and air from the hospital pressurized gas system require humidification, while those operating with pressurized oxygen in combination with room air partially rely on the humidity of room air. These considerations must be contextualized with respect to ideal environmental conditions for human breathing, including open-air environment, high atmospheric pressure, and a temperature of 20 °C (40% relative humidity) [24]. The good practice for non-invasive ventilation recommends active gas humidification, particularly when the inspiratory fraction of oxygen (FiO_2_) exceeds 60%. However, establishing a standard FiO_2_ threshold is challenging due to variations in room air humidity in hospital units. Guidelines recommend active humidification with a relative humidity of 40% at 28 °C, corresponding to an absolute humidity of 10 mg/L [13,14,15].

## 3. Technical Aspects

### 3.1. Passive Humidification

Passive humidification is achieved through the use of heat and moisture exchanger (HME) devices. These devices capture water vapor perspired from the lungs during the expiratory phase and reintroduce heat and moisture during the subsequent inspiration. Furthermore, these filters also exhibit antibacterial properties and provide isolation for the ventilatory circuit. Their performance varies according to the model, achieving a maximum absolute humidity value within the filter approximately of 30–32 mg/L at temperatures ranging from 27–30 °C and relative humidity levels of 90–95% [19,25,26,27]. HME devices can be classified into hydrophobic and hygroscopic types [28].

Hydrophobic HME devices feature a membrane with pores measuring 0.2 μm in diameter. These pores enable the passage of gas and water vapor while preventing the passage of suspended water droplets. Inside the HME, the porous membrane is positioned on the ventilator circuit side, and the entire volume of the filter is occupied by a condensation surface [29]. This condensation cell, like the first airways, enables the capture of a portion of exhaled water vapor. Its performance improves with an increased temperature difference between the patient’s side and the ventilator side. Under standard temperature conditions, the temperature of the gas exhaled by the patient is approximately 33 °C, featuring an absolute humidity of 36 mg/L and 100% relative humidity. Following the filter, the gas temperature decreases to 20 °C, resulting in an absolute humidity of 18 mg/L. Therefore, the filter captures and returns 18 mg/L of water and heat. During the subsequent inspiration, the temperature and moisture contained in the HME are added to those originating from the ventilator medical gases (22 °C of temperature and 0 mg/L of absolute humidity) and the additional 8 mg/L contributed by the evaporation of condensate on the post-filter tube [1]. This cumulative effect results in reaching a temperature of 35 °C with an absolute humidity of 26 mg/L, equivalent to the desired conditions at the tracheal carina. At that point, the lower airway conditions the pre-alveolar air to the physiological temperature of approximately 37 °C, featuring an absolute humidity of 44 mg/L and reaching 100% relative humidity (Figure 2) [1,17,26,27].

Hygroscopic HMEs function similarly to hydrophobic ones, but they include an additional layer of hygroscopic material, typically calcium or lithium-based. This additional layer enhances the storage capacity for both water and heat. These filters are the most effective in passive humidification systems and can retain approximately 28 mg/L of absolute humidity, which is 10 mg/L more than hydrophobic ones. This capability brings the values at the tracheal carina at about 35 °C with 36 mg/L of water vapor (Figure 2) [26,27,29].

Many HMEs also possess antibacterial and antiviral capabilities, making them HME-Filters with a filtration efficiency of approximately 99.9%. Conversely, certain filters are exclusively designed for antibacterial and antiviral activity without providing heat and moisture exchange. Therefore, careful consideration is essential when selecting filters, considering their intended purpose and position in the circuit [29].

Finally, active HMEs are humidifiers positioned between the HME filter and the endotracheal tube, incorporating an electrical heating ceramic element that vaporizes water in the airway. These systems provide an absolute humidity approximately 3–4 mg/L higher than the passive HME action, but their actual clinical benefits are not yet fully proven [25,30,31,32].

HMEs must be replaced every 24 h, or according to the manufacturer’s instructions, otherwise they may have a decline in their specific performance, potentially leading to increased resistance to ventilation flow. It is important to note that all certified HMEs must adhere to international performance recommendations, such as the European Recommendations (ISO 9360/1992(E)). These guidelines encompass criteria such as limited resistance (drop in inspiratory flow below 25 L/min at 30 cmH_2_O pressure, a pressure drop less than 5 cmH_2_O at 60 L/min of flow), a volume less than 50 mL, and proper humidification within tidal volumes ranging from 200 to 1000 mL [33].

### 3.2. Active Humidification

Active humidification is achieved using specialized equipment designed to actively supply heat and moisture to medical gases [34]. Various active humidification systems, also known as heated humidifiers (HHs), have been developed and commercialized over the years. Four main types can be distinguished [35]:Bubble-through humidifiers: gases pass through a heated water reservoir where they are humidified through bubbling.Passover humidifiers: gases are humidified by passing through heated cells equipped with permeable membranes or water-free surfaces.Counter flow humidifiers: water is heated outside the system and then flows within the ventilatory circuit, counter to the direction of gases, providing humidification.Inline vaporizer humidifiers: gases are humidified through a process of direct water vaporization inside the ventilatory circuit.

#### Passover Humidifier

Passover heated humidifiers (HHs) are the most used devices in the ICU, primarily due to their favorable performance-to-cost ratio [36,37,38]. Passover operates based on the principle of regulating the temperature gradient within the ventilatory circuit. Two distinct models are available: the “Membrane” model and the “Free Water” model, depicted in Figure 3 and Figure 4, respectively. Both systems are closed systems, connected to the ventilator through dedicated circuits. These circuits have the fundamental characteristic of having a heated inspiratory tube branch [39].

The passover membrane humidifiers includes a metallic cylinder equipped with a permeable porous membrane connected to an external water reservoir, enabling a continuous imbibition of the membrane [39].

Free water humidifiers, on the other hand, are composed of a bell with a metallic plate at the bottom [40]. In this case, water drops passively, filling the bottom of the bell. In both systems, heating is provided by an electrical resistance directly in contact with the metallic parts. The variables that regulate the gas humidification process include the water/air contact surface, the gas flow rate, and the temperature. Among these variables, only the temperature regulation becomes the parameter available to manipulate the process. A higher water temperature results in a proportional increase in evaporation and absolute humidity. This parameter, therefore, determines the actual amount of water input into the system. After achieving this, it is crucial to ensure that evaporated water reaches the patient without condensation. This is facilitated by the heated inspiratory limb of the ventilator circuit. This feature is important to prevent humidity from condensing on the cold surfaces of the tube.

Active electrical pass-over humidifiers have three main modes of regulation:Invasive mode and non-invasive mode. This parameter allows for the adjustment of the working temperature range and gas absolute humidity, depending on whether the upper airways are bypassed or not. As a reference, depending on the humidifier models, the temperature level allowed in the invasive setting typically ranges from 33 °C to 39 °C, while for the non-invasive setting, it ranges between 28 °C and 37 °C [41].Patient temperature. This parameter allows for setting the desired gas temperature at the circuit “Y” point, which is the bifurcation between the inspiratory and expiratory branches [42].Temperature gradient. With the presence of two temperature sensors, the inspiratory heated branch tube enables the setting of a temperature difference between the patient’s temperature and the evaporation cell temperature. Temperature gradients typically range between −3 and +3 °C. Consequently, this parameter indirectly allows the setting of the temperature of the water in the heating cell, where a higher cell temperature corresponds to increased water evaporation (absolute humidity) (Figure 5) [42].

Therefore, these settings influence both absolute humidity (the amount of water input into the system) and the gas’s ability to retain water vapor (actual water amount reaching the patient) [43]. As an example, if the patient temperature is set to 35 °C, and the gradient is set to +2 (positive gradient), this will result in a warmer cell (37 °C), leading to an increase in absolute humidity due to greater evaporation. Conversely, this setting will result in a reduction in absolute humidity for the patient because the gradual cooling of the gas along the inspiratory branch will lead to significant water condensation [39]. However, the more water lost in condensation on the tube, the lower the actual humidity reaching the patient. On the contrary, a negative gradient will result in a colder cell, leading to less evaporation and consequently lower absolute humidity [44]. However, the gradual gas heating will maintain the water “bioavailability” to the patient because an increase in temperature leads to a decrease in relative humidity, implying a greater capacity of the gas to transport water vapor [26].

In Figure 6, the operational algorithm for setting up a passover humidifier is depicted.

In summary, the key features of a passover-heated humidifier should include: a broad range of selectable temperature settings, easily adjustable through various ventilation modes; consistent performance even at high flow rates; an effective water sampling system; and a display providing feedback on the patient’s temperature and the evaporation cell status.

## 4. Settings

### 4.1. Best Setting for Invasive Ventilation

Set invasive mode.Set patient temperature to 37 °C (±2): this setting allows an adequate pre-alveolar physiologic gas temperature to be reached with 44 mg/L of absolute humidity and 100% relative humidity [45].Gradient settings:A. Zero Gradient: the goal is to achieve a balanced humidity relationship. Although theoretically maintaining a constant humidity, it is necessary to be cautious of suboptimal tube performance and room temperature influence. This setting is recommended when the inspiratory branch has condensation collectors.B. Negative gradient (−1 or −2): this setting minimizes condensation risk, albeit with a slightly lower humidification level, aligning with ventilator-associated pneumonia prevention. This setting is recommended when the inspiratory branch lacks condensation collectors [45].Optimal setting for ICU patients: adjusts patient humidification by regulating temperature rather than changing the gradient. Ensure patient temperature within 37 °C ± 2 (35 °C to 39 °C) for correct absolute humidity [17,46].Thermoregulation considerations: use caution when considering extreme measures to decrease the patient body temperature through humidifier temperature adjustments. Instead, it is advisable actively warm inspired gases for hypothermia management, aligning with physical treatments [47,48].

### 4.2. Best Setting for Non-Invasive Ventilation

Set non-invasive mode.Set patient temperature to 28 °C: this temperature facilitates an evaporation of 10 mg/L with 40% of relative humidity [13,15].Gradient setting: zero gradient. This setting ensures an optimal balance between provided humidity and the humidity reaching the patient. Given the reduced humidity production at 28 °C, this setting prevents condensation in the tube or in the ventilatory device.Special consideration during High Flow Nasal Cannula (HFNC) therapy:A.During high flow nasal cannula therapy, it is necessary to elevate the humidifier temperature due to the direct high flow in the patient’s upper airways.B.Set the patient target temperature at 32 °C. The temperature setting can be adjusted within a range of 30–34 °C depending on the patient’s comfort [49,50,51,52].

## 5. Clinical Evidence and Future Directions

### 5.1. Clinical Evidence

The optimal heating and humidification of medical gases are pivotal aspects of respiratory care, particularly during mechanical ventilation. As outlined previously, inadequacies in gas conditioning can lead to a spectrum of complications, ranging from mucosal injuries to compromised surfactant functionality. Achieving the delicate balance between delivering sufficient moisture and preventing thermal injury remains a clinical challenge [12,53,54,55]. Advancements in humidification technology have significantly influenced contemporary clinical practice. The transition from aerosol-based devices to active and passive humidification systems reflects ongoing efforts to enhance patient outcomes and streamline respiratory therapy protocols. The emergence of passive humidification devices, such as heat and moisture exchangers (HMEs) underscore the importance of maintaining moisture levels while minimizing circuit-related complications [33,56,57,58]. Moreover, distinguishing between invasive and non-invasive ventilation modalities is paramount when configuring humidification parameters. While invasive ventilation necessitates the meticulous control of temperature and humidity to mitigate airway complications, non-invasive ventilation presents unique challenges due to varying gas sources and treatment durations.

The recommended target values for absolute humidity serve as guiding principles in optimizing gas conditioning across different ventilation modalities [59,60,61,62]. Understanding the interplay between temperature, humidity, and patient comfort is essential for refining humidification protocols in clinical settings.

### 5.2. Future Directions

Ongoing research endeavors are warranted to explore novel approaches, such as active humidification systems, and to elucidate their clinical efficacy and cost-effectiveness. Collaborative efforts among clinicians, engineers, and researchers will drive innovation and foster the development of tailored solutions to meet the evolving needs of ventilated patients [60,61,62,63]. The limitations associated with humidification strategies are manifested by the variability in patient physiology, environmental conditions, and device performance, emphasizing the necessity for individualized approaches to gas conditioning [64,65,66]. Despite advances in humidification technology, several areas warrant further investigation. Future research efforts should focus on optimizing humidification strategies for specific patient populations, evaluating the long-term effects of humidification on airway integrity and respiratory outcomes, and developing innovative approaches to enhance humidification efficiency and patient comfort. Collaborative initiatives involving clinicians, researchers, and industry stakeholders are essential for advancing the field of respiratory care and improving outcomes for ventilated patients.

## 6. Conclusions

In conclusion, the integration of heating and humidification in medical gas management represents a cornerstone of modern respiratory care. Moving forward, continued research endeavors and technological innovations will further refine our understanding and implementation of optimal gas conditioning strategies, ultimately enhancing patient outcomes and quality of care.

## Figures and Tables

**Figure 1 medsci-12-00026-f001:**
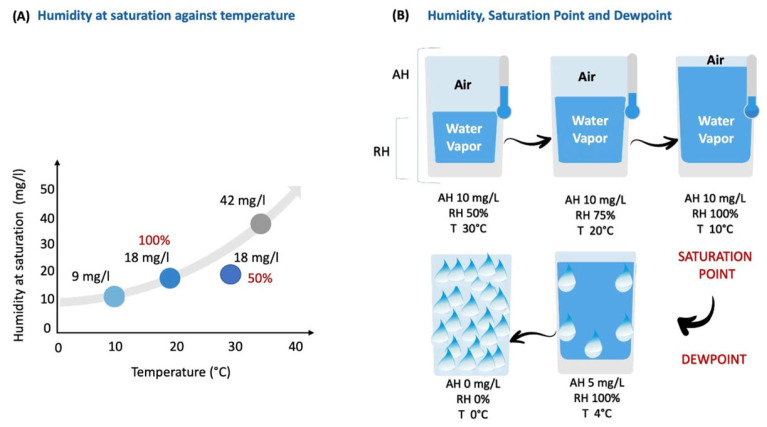
The direct impact of temperature on the air’s ability to carry water vapor. For details, see text below. AH, absolute humidity; RH, relative humidity; T, temperature.

**Figure 2 medsci-12-00026-f002:**
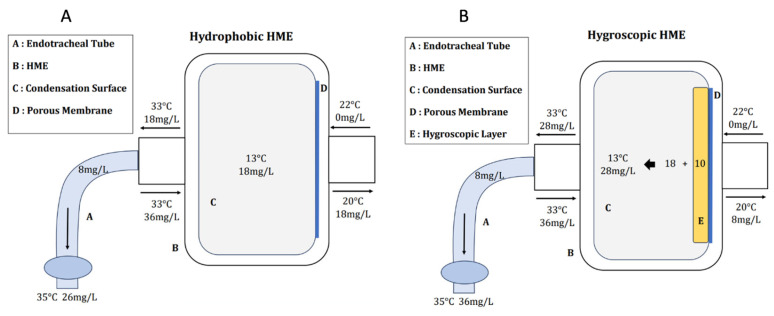
Hydrophobic HME (Panel (**A**)) and hygroscopic HME (Panel (**B**)). HME, heat moisture exchanger.

**Figure 3 medsci-12-00026-f003:**
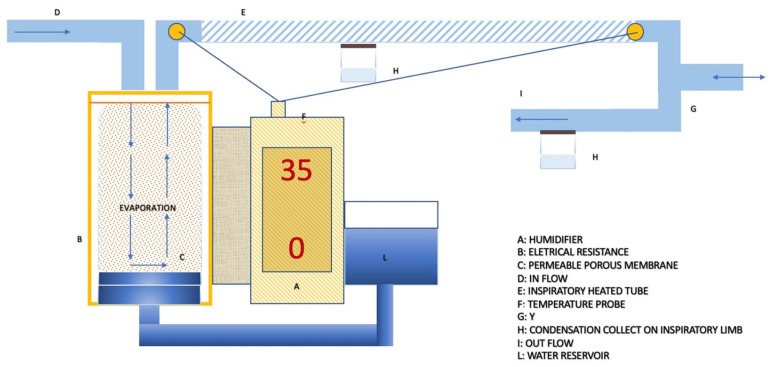
The passover membrane humidifier is integrated into an older ventilatory circuit with condensation collected on both ventilation limbs. This configuration allows for a humidifier setting at 35 °C with zero gradient.

**Figure 4 medsci-12-00026-f004:**
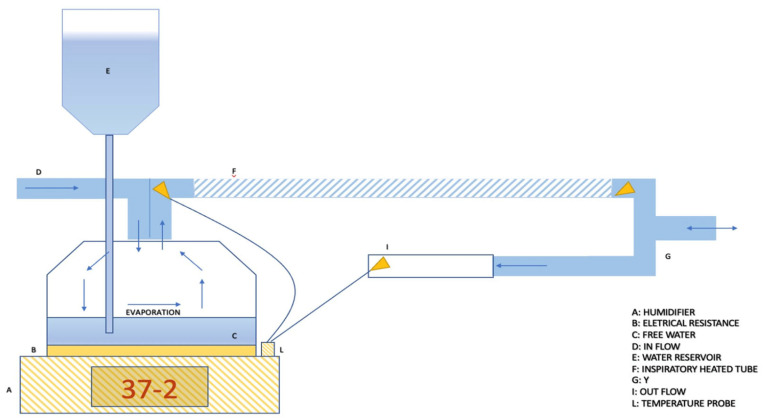
The passover free water humidifier is integrated into a ventilatory circuit without condensation collection. This is possible thank to the high-performance heated inspiratory and expiratory tubes. In this configuration, a humidifier setting of 37 °C with a negative gradient is required.

**Figure 5 medsci-12-00026-f005:**
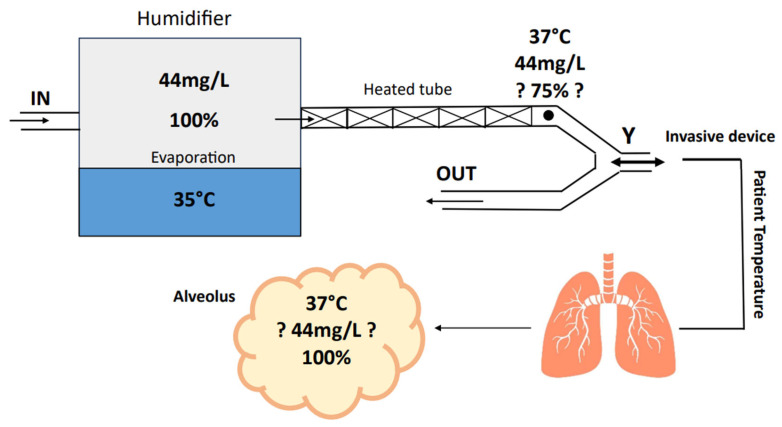
Absolute humidity (mg/L), Temperature (°C) and relative humidity (%) in ventilatory circuit. Heated tubes reduce relative humidity while maintaining the absolute humidity. This allows for an additional increase in humidity from the natural patient airways.

**Figure 6 medsci-12-00026-f006:**
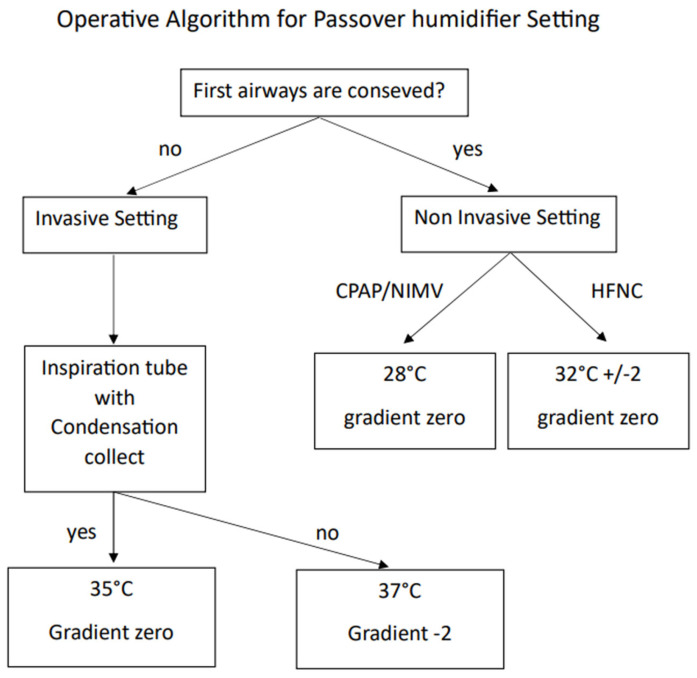
Operative algorithm to set up a passover humidifier.

## Data Availability

No new data were created or analyzed in this study. Data sharing is not applicable to this article.

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
