# Peer review of "Humidification during Invasive and Non-Invasive Ventilation: A Starting Tool Kit for Correct Setting"

_medsci, 2024, doi:10.3390/medsci12020026_

Round 1

Reviewer 1 Report

Comments and Suggestions for Authors

Review article "Humidification during invasive and non-invasive ventilation: a starting tool kit for correct setting" is well written review article on the topic that is very important, but not so well presented, for majority of ICU patients. I can strongly recommend this manuscript to be published without corrections.

Author Response

We thank the reviewer for the comment.

We are very pleased that the manuscript was appreciated.

Reviewer 2 Report

Comments and Suggestions for Authors

These comments accompany the manuscript "Humidification during invasive and non-invasive ventilation: a starting tool kit for correct setting." This review was well written and contains very clear and helpful figures.

The only recommendation that I have for the authors is to add paragraphs to section 5. It is a bit difficult to digest all of the information as it is written. Breaking the ideas up will help key points be more obvious to the average reader.

Author Response

We thank the reviewer for this suggestion.

We have divided section 5 into two paragraphs (clinical evidences and future directions) in order to make the concepts expressed clearer and simpler.

The change on the manuscript is highlighted in yellow.

Reviewer 3 Report

Comments and Suggestions for Authors

This manuscript describes very impartant topic. The manuscript is well prepared, organized and covers the topic well.

I have several suggestion of minor modification of the manuscript.

Selection of HME is always a tradeoff between dead space, resistance and quality of humidification - i would appreciate authors comment on these topic or suggestion which parameter is the most important (based on patient population)

No all system for active humidification allow setting of all suggested parameters. Would it be possible to add some table with ideal properties of active humidifier

Authors mention the reccomended temperature setting for NIV. Are there any data on NIV tollerance or failure rate with different temperature ranges.

Authors also mention that ventilators that use their own turbine need lower addition of water to appropriately humidify used mixture of air and oxygen. Does it has any practical consequence?
